# 'You feel like you've been duped': is the current system for health professionals declaring potential conflicts of interest in the UK fit for purpose? A mixed methods study

Margaret McCartney [1], Katrin Metsis [1], Ronald MacDonald [1,2], Frank Sullivan [1,3], Gozde Ozakinci [4], Anne-Marie Boylan [5]

[1]School of Medicine, University of St Andrews, St Andrews, UK
[2]University of Dundee School of Medicine, Dundee, UK
[3]North York General hospital, Toronto, Ontario, Canada
[4]Department of Psychology, University of Stirling, Stirling, UK
[5]Nuffield Department of Primary Care Health Sciences, Oxford University, Oxford, UK

**Correspondence to**
Dr Margaret McCartney;
mm494@st-andrews.ac.uk

## ABSTRACT:

**Objective** To understand: if professionals, citizens and patients can locate UK healthcare professionals' statements of declarations of interests, and what citizens understand by these.

**Design** The study sample included two groups of participants in three phases. First, healthcare professionals working in the public domain (health professional participants, HPP) were invited to participate. Their conflicts and declarations of interest were searched for in publicly available data, which the HPP checked and confirmed as the 'gold standard'. In the second phase, laypeople, other healthcare professionals and healthcare students were invited to complete three online tasks. The first task was a questionnaire about their own demographics. The second task was questions about doctors' conflicts of interest in clinical vignette scenarios. The third task was a request for each participant to locate and describe the declarations of interest of one of the named healthcare professionals identified in the first phase, randomly assigned. At the end of this task, all lay participants were asked to indicate willingness to be interviewed at a later date. In the third phase, each lay respondent who was willing to be contacted was invited to a qualitative interview to obtain their views on the conflicts and declaration of interest they found and their meaning.

**Setting** Online, based in the UK.

**Participants** 13 public-facing health professionals, 379 participants (healthcare professionals, students and laypeople), 21 lay interviewees.

**Outcome measures** (1) Participants' level of trust in professionals with variable conflicts of interest, as expressed in vignettes, (2) participants' ability to locate the declarations of interest of a given well-known healthcare professional and (3) laypeoples' understanding of healthcare professionals declarations and conflicts of interest.

**Results** In the first phase, 13 health professionals (HPP) participated and agreed on a 'gold standard' of their declarations. In the second phase, 379 citizens, patients, other healthcare professionals and students participated. Not all completed all aspects of the research. 85% of participants thought that knowing about professional declarations was definitely or probably important, but 76.8% were not confident they had found all relevant information after searching. As conflicts of interest increased in the vignettes, participants trusted doctors less. Least trust was associated with doctors who had not disclosed their conflicts of interest. 297 participants agreed to search for the HPP 'gold standard' declaration of interest, and 169 reported some data. Of those reporting any findings, 61 (36%) located a relevant link to some information deemed fit for purpose, and 5 (3%) participants found all the information contained in the 'gold standard'. In the third phase, qualitative interviews with 21 participants highlighted the importance of transparency but raised serious concerns about how useful declarations were in their current format, and whether they could improve patient care. Unintended consequences, such as the burden for patients and professionals to use declarations were identified, with participants additionally expressing concerns about professional bias and a lack of insight over conflicts. Suggestions for improvements included better regulation and organisation, but also second opinions and independent advice where conflicts of interest were suspected.

**Conclusion** Declarations of interest are important and conflicts of interest concern patients and professionals, particularly in regard to trust in decision-making. If declarations, as currently made, are intended to improve

## STRENGTHS AND LIMITATIONS OF THIS STUDY

⇒ This is the first study we are aware of to test whether professionals and laypeople can find and interpret declarations of interest made by professionals in the UK.
⇒ Co-designed with a patient panel.
⇒ Pragmatic, real-world study design.
⇒ Participants were likely to be atypical and not representative of most professionals or patients.
⇒ Success in finding declarations was judged even if declarations were incomplete, overestimating the availability of declarations.
⇒ Declarations are not of equal importance but were graded equally for the purposes of analysis.

transparency, they do not achieve this, due to difficulties in locating and interpreting them. Unintended consequences may arise if transparency alone is assumed to provide management of conflicts. Increased trust resulting from transparency may be misplaced, given the evidence on the hazards associated with conflicts of interest. Clarity about the purposes of transparency is required. Future policies may be more successful if focused on reducing the potential for negative impacts of conflicts of interest, rather than relying on individuals to locate declarations and interpret them.

**Trial registration number**  The protocol was pre-registered at https://osf.io/e7gtq.

## INTRODUCTION

Contention has surrounded the definition, declaration and management of interests in medical practice for decades. When the US Congress asked 'Is science for sale?' in a series of hearings regarding research fraud in the late 1990s, declarations of interest were generally haphazard and voluntary. In 2009, The International Committee of Medical Journal Editors produced a standardised template for recording and publishing declarations, which has been serially upgraded since.[1] However, declaring interests has subsequently become an expectation in routine clinical practice, not just academic publishing, and is now law via Sunshine Acts for healthcare professionals in several countries. In National Health Service (NHS) England, guidance published in 2017 stipulates how and which declarations should be publicly made.[2]

There is evidence of widespread harm to patients and healthcare systems because of financial conflicts of interest (COI). For example, guidelines dealing with opioid prescribing for non-cancer pain (recognised as driving 'the opioid crisis') have had a 'pervasive presence' of influence of the pharmaceutical industry among guideline authors or organisations receiving funding from them.[3] Doctors with financial COI are more likely to have favourable views on side effects from medications.[4] Meanwhile, doctors receiving gifts and education from pharmaceutical sales representatives have poorer quality prescribing and believe their peers, and not themselves, are adversely affected by conflicts.[5]

Healthcare professionals are often asked to make declarations, or 'declarations of interest' (DOI) in different venues, for example, workplaces, academic journals and conference presentations. Such declarations relate to the act of recording any interests which may be, or could potentially become a conflict, depending on context or circumstances. A conflict of interest, however, is 'a set of circumstances that create a risk that an individual's ability to apply judgement or act in one role is, or could be, impaired or influenced by a secondary interest'.[6] The UK government commissioned the Independent Medicines and Medical Devices Safety Review (IMMDS) in 2020, to investigate how the health service responded to concerns about medical products, and one conclusion was that serious problems are caused by a lack of transparency of COI.[7] However, there may be uncertainty over when a declaration represents a conflict, particularly when a declaration is prepared in advance and used for multiple purposes. The review recommended that doctors should make statutory disclosures of interests on a central register, including, for example, consultancy payments from pharmaceutical companies, patents for technological devices or shareholdings in device companies. The General Medical Council already recommends that doctors are open and transparent about their interests. Previous inquiries, for example, the Health Select Committee in the 2005 report 'The Influence of the Pharmaceutical Industry', recommended that all healthcare professionals make statutory DOI to their regulator,[8] but this did not occur.

There are multiple types of potential COI. NHS England classes these as financial or non-financial. The first is where an individual receives or may receive a direct financial benefit from the consequences of the awarding of funding.[2] This can be direct (eg, a shareholder receiving more funds for increased sales in a product) or indirect (where a person closely associated, such as a spouse or business partner will benefit similarly). This may also include industry funding to attend conferences, to advise or provide consultancy, fees for speaking or research funding. Non-financial interests can be professional (eg, a decision likely to enhance a career or status, or an intellectual bias) or non-professional (where other interests, such being a member of a lobby group, may compete).

Many countries mandate disclosures of interest by health professionals[9]. The US Sunshine Act, which mandates transparency of payments from industry, was implemented in 2013. The same year, a Disclosure Code by the European Federation of Pharmaceutical Industries and Associations was implemented to improve transparency, but is variable by country where it conflicts with other laws.[10] In the USA, the Act does not seem to have resulted in improved patient knowledge of physician payments, with 3% of people knowing whether their own doctor has received industry payments.[11] DOI may serve multiple purposes. For example, in guideline committees,[12] individuals may only be allowed limited roles or excluded entirely on the basis of conflicts and declarations are used to judge this. Regulators may stipulate specific conflicts which must be declared to patients.[13] Healthcare professionals may also use COI statements to make judgements about the validity of opinions or published research findings. Managers may have to use declarations to ensure that commissioning or procurement decisions are made by non-conflicted individuals. The use of declarations by patients has been emphasised in the IMMDS review, which has said "we deserve to know" with a patient stating the need to access declarations in order to 'reach informed decisions about who is best to treat us'.

Disclosures of interest have been historically acknowledged as necessary but also with multiple inadequacies in practice, including poor quality recording and subsequent management.[14] There are additional concerns that

statutory declarations, rather than reducing the negative impact of COI, may transfer the burden of searching for, interpreting and acting on them to patients.[15] Further, there are concerns that disclosing interests may increase bias through 'moral licence', when doctors believe that disclosure equals management of conflicts, and where patients and citizens believe that transparency negates bias.[16] In the USA, patients who are explicitly told about a doctor's financial conflicts with industry do not appear to change their attendance with the doctor, with no loss of trust[17]; however other research suggests that disclosure may result in an increase in trust.[18] Patients at US cancer centres have low levels of concern about doctors' financial conflicts from pharmaceutical companies[19]; however, this study did not ask unconcerned patients what they thought the negative impacts might be. The UK has a voluntary system, Disclosure UK, where payments to professionals from pharmaceutical companies are published annually, but most money is undeclared.[20] NHS Trusts in England mandate disclosures for staff but these are incomplete and of poor quality.[21] Therefore, while voluntary and mandated disclosures from healthcare staff are available to patients in the UK, these are problematic due to their quality, and it is unknown whether these incur unintended hazards from disclosures, and whether they enable evidence-informed decision-making.

As the NHS responds to the Cumberlege report,[22] potential actions in the recording and managing of DOI require consideration. Little is known about how much knowledge patients, citizens, professionals and policymakers have regarding COIs among healthcare professionals, or locating and interpreting this information. A 2016 systematic review on knowledge, beliefs and attitudes of patients and the public towards interactions between professionals and the pharmaceutical and device industry found low levels of concern about financial conflicts but did not include any studies from the UK or Europe, with the exception of Turkey.[23]

There is a large gap in our understanding of the most effective way to declare and interpret interests, and what patients, citizens and professionals think about the impact of them. This is required in order to ensure that any change in policies are beneficial and meet their intended purpose. The aim of the study was to understand: if professionals, citizens and patients can locate UK healthcare professionals' statements of DOIs, and what citizens understand by these.

## METHODS
### Patient involvement
We thank the lay group for their advice before, during and after the study. They helped to design the methods, the questionnaires, the choice of professional participants and assisted in recruiting lay participants. The results will be shared with them.

This research takes a pragmatist perspective.[24 25] While the different epistemological standpoints of qualitative and quantitative methodologies are acknowledged, a mixed methods approach was chosen to gain a comprehensive understanding of the citizen and patient participants' perspectives of COIs in health professionals. The online survey, completed in the second phase, provides a quantified understanding of the respondents' perceptions of health professionals' variable interests and the process of locating declarations. Qualitative interviews in the third phase investigates lay perspectives of declarations and COI in healthcare, and current declaration strategies. Thus, quantitative and qualitative data are used to gain an understanding of the different aspects of the phenomenon, which are integrated but keeping their epistemological differences.[26]

### Phase 1: methods development
The research team established a Public and Patient Involvement (PPI) group. This consisted of four laypeople who are involved with patient representation at the UK Royal College of General Practitioners or who have been involved in lay activism regarding surgical mesh. The PPI group helped design the methods, the questionnaires, the choice of professional participants and assisted in recruiting lay participants.

Professionals (HPP, health professional participants) were identified and invited on the basis of their recent history of giving medical advice or information to the public, where a citizen might reasonably wish to know their DOI. A mixture of healthcare officials based in the UK (eg, doctors employed by the government), heads of royal colleges/professional societies, pressure groups, NICE (National Institute for Health and Care Excellence) and Scottish Intercollegiate Guidelines Network (SIGN) authors and doctor journalists were invited. This was done in batches of 10–15 aiming to recruit 12–15 in total. A standard process was used to search for the professionals' COI (online supplemental appendix 1). Each professional was asked to check the findings and approve or disagree with them. This formed the 'gold standard'. Statements within each 'gold standard' declaration were divided into 'financial professional', 'non-financial professional', 'personal' and 'indirect' as per the NHS England disclosure framework.[2] This recruitment of HPP completed the first phase of the study.

### Phase 2: online survey
An online survey (using Qualtrics) was developed and tested with the lay group . This contained four scenarios explaining and asking opinions on potential financial COI. We described these to participants as 'interests' and not 'conflicts'. A wide range of participants from citizen, patient, activist, professional and student groups were formally invited (online supplemental appendix 2) to participate online, with patient and professional networks asked to publicise the research via social media. The invitation contained a web link to the survey which prospective participants could click on. Demographic and occupational information was requested.

Healthcare professionals were additionally asked how many DOI forms they were asked to complete a year. Participants were then asked to express their trust in a doctor, and whether they felt they were acting in their best interests, within a vignette about a surgical procedure, where doctors making a recommendation had variable but increasing financial interests. These were either not present, present and declared or not declared but found by the participant. This was designed to give information about what a DOI is, to gauge views on their importance in terms of impact on personal decision-making and to explain and prime participants as to what a DOI was before being asked to locate them (online supplemental appendix 3). These used 5-point Likert scales.

The participants were then randomly assigned to one HPP from the group of 13, with a brief introduction about the person, and asked to spend around 10 min (as suggested by the PPI group) to search for and report their DOI. However, participants could spend as little or as much time as wished. We explained that we were not seeking any 'personal' information such as age or marital status or address but either a statement of DOIs, or the interests they found. Information on how long participants felt reasonable to search for a conflict was also sought.

They were asked to report findings in a web form. Excel was used to tabulate each piece of data reported by participants. These were compared with the professional's 'gold standard', which were divided into financial and non-financial declarations. Each was deemed 'significant' or 'non significant' (online supplemental appendix 4) based on NHS England criteria.[2]

Participants' responses to the HPP 'gold standard' were analysed by hand. Participants were asked to record all information found, supplying relevant web links. The responses were recorded and assessed for accuracy and completeness via comparison with the 'gold standard' (MMC and RM). Given the difficulty the task was expected to pose, marking was generous. We did not ask either the professionals or the participants to categorise interests within each NHS England category of declarations (eg, financial, non-financial professional, non-financial personal interests and indirect interests) but the research team categorised and marked them on this basis. If a 'gold standard' declaration containing full details was not available online for participants to locate, locating a single declaration from each category of declaration was marked as successful, even if incomplete. 10% of the results were checked by the second researcher for accuracy and no disagreements were found. Null declarations were excluded in the tally (i.e., the denominator was according to the declarations present). This strategy would have overestimated the tally of declarations found, by design. At the end of the survey, all laypeople were asked if they would like to take part in a qualitative interview at a later stage.

### Third phase: interviews

In the third phase, all lay participants who stated they were willing to take part in an interview were contacted. This was a semi-structured, qualitative interview where their views about both DOI and where these represented conflicts could be explored. Fifty agreed to be contacted and of these, 21 consented to be interviewed (KM and MMC) (online supplemental appendix table 1). A semi-structured interview format was chosen as this method is well suited for the exploration of opinions and perceptions, enabling the further exploration of the topics identified by respondents. A topic guide was developed for use in the interviews with the understanding that it would be iteratively refined and new questions added as data collection progressed. Questions included exploring what DOI were, and their purpose, what a 'conflict of interest' in healthcare professionals means, what was known about them, if they were perceived as important and how they should be managed. The initial topics were developed after discussion with the lay group and related to real-life practice in the UK, where the medical regulator advises that professionals should use self-judgement to decide when a conflict occurs and when it should be declared.[13] All interviews were transcribed verbatim and transcripts were transferred to NVivo for analysis.

Interview data was analysed using thematic analysis.[27] KM completed 20 out of 21 interviews (MMC did one). KM created the initial set of codes based on six transcripts. MMC read and discussed all transcripts, interview notes and initial codes. This was then discussed with A-MB alongside, with conceptualisation of potential themes. Next, all transcripts were coded in NVivo and initial themes were developed by KM. Initial themes were further developed and refined with input from MMC and A-MB. KM, MMC and A-MB developed the final three themes and these were agreed by the whole team.

### RESULTS
#### Phase 1

A total of 65 professionals were invited; 3 sent a formal declination and 13 consented to take part. Two did not respond after further information was requested and sent; the remainder did not reply. The research team completed the 'gold standard' and presented it to the HPP. All but one statement was agreed for one HPP. After discussion it was deemed inaccurate by both the professional involved and the research team. No information contained within the 'gold standards' created was not available online.

The professional participants' declarations varied markedly, with some having minimal or only professional declarations to make (minimum 6, including job roles) and others having multiple financial declarations (maximum 20, eg, sponsorship, consultancy, shares, private practice, patents, multiple professional roles). Within NHS England categories of declarations (financial, non-financial professional, non-financial personal

and indirect) some participants had multiple in one category and others were blank.

## Phase 2
### Survey findings
In the second phase, 378 individuals participated and answered at least some of the questions. Among them 266 (70%) were women and the vast majority lived in the UK (354, 93%). Also, 141 (37%) described themselves as lay (citizens/patients), and the remainder as either training, trained or working in healthcare. The median age was 50–59 (range 18–80+). The majority of healthcare professionals participating were general practitioners (GPs) (59, 25% of healthcare participants) followed by physicians and nurses. Of 230 healthcare professionals, 41% (95) filled in between 1 and 4 DOI forms per year, with 35% (81) filling in none, and 23% (94) completing more than 5 (online supplemental appendix 5). A few participants did not complete all parts of the survey, meaning that some totals vary.

In the vignettes, when no interests were actively declared, 95% (335) of participants trusted the doctor either moderately, a lot or completely and the same number felt they were acting in their best interests. If the doctor declared that they had been sponsored to travel to a conference by the company making a recommended joint replacement, the trust scored at moderately, a lot or completely fell to 79% (184) with 83% (286) judging as acting in their best interests. If the doctor declared heavier financial interests, including a patent and shares in the company, 54% (187) trusted the doctor moderately, a lot or completely, with a similar number (53%, 184) believing the doctor was acting in their best interests. When the doctor did not declare any interests, but was subsequently found by the participant to be acting as a consultant to the company, 22% (74) trusted the doctor moderately, a lot or completely, and 24% (80) felt they were acting in their best interests. There was a clear progression of decreased trust and decreased belief that the doctor was acting in their best interests with increasing financial interests. The least trust-generating scenario was where the conflict was not directly disclosed (online supplemental appendices 6 and 7).

Survey respondents were then asked if they wished to continue to the task of searching for an individual's DOI. Among respondents 297 participants responded positively, 169 reported some data and 128 reported no findings. Each participants' findings were individually assessed to determine the proportion of declarations located within each type of category of interest, as described in the HPPs' 'gold standard' (online supplemental appendix 8). Sixty-one found a source designed as a formal DOI, for example, on guideline committee websites, which should have made it 'fit for purpose', containing at least some of the information from each of the four categories of NHS England guidance on declarations (financial professional and non-profession, non-financial professional and non-professional).

However, these formal, online declarations, despite being designed to be 'gold standard', were not all complete when compared with the standard we had generated with the HPP. Five participants were able to replicate the 'gold standard' in their search. The top five highest cited links for the amount of accurate declarations were held in an organised register, such as NICE, parliamentary register, government website or whopaysthisdoctor.org. When asked how long it should take to find a healthcare professional's DOIs, participants gave a range of 0–120 mins, mean of 8.63 min.

Participants were asked how easy or difficult they found the search. Of 212 answering, 67.5% (143) said it was extremely or very difficult, and 36 (16.9%) somewhat or extremely easy. While 76.8% (163) were not confident they had found all necessary information; 86% (184/213) said that finding DOI were definitely or probably important.

### Qualitative interview findings
At the end of the survey, all 50 laypeople who consented were invited for an interview. All were individually contacted. Twenty-one patients/citizens agreed to take part in this third phase and are referred to as they self-described. Thirteen were women and eight men; four described themselves as activists, and six as patients, with a variety of backgrounds including working in the pharmaceutical industry, research or the public sector. The age range was 18–80, mean of 62. Interviews lasted between 19 and 51 min and were transcribed verbatim for analysis. Participants could review their transcript. Nine requested them and three returned them with further comments; one corrected minor inaccuracies and the others added further comments.

Four themes were developed: *COIs in healthcare are difficult to define; DOI can be hard to find; COIs may present both challenges and benefits for patients; and COIs need to be carefully managed.*

Additional quotations are included in online supplemental appendix 9. Demographic information on participants is contained in online supplemental appendix table 1.

### COIs in healthcare can be difficult to define
Consistently, COIs were described as situations where care and treatment decisions benefitted the professional before the patient. However, although participants described a wide range and descriptions of COIs, they agreed that the concept and definition could be difficult to define. Some participants had acquired an understanding of COIs and declarations from different roles, including patient representative roles and experience of working in research or industry.

> I know very little about it…. It makes me think of when drug companies go round, and they do a lunch for the doctors and they do a presentation about their product and the doctors prescribe their product. But I feel it's much wider and there's lot of other circumstances. I probably don't know much more other than a vague awareness that there are these other circumstances where there are potentially conflicts

of interest. I don't really know much about them. (P1, patient)

Perceptions of a COI included a variety of financial COIs, particularly from industry. Examples of non-financial COIs given included gifts from pharmaceutical companies, Healthcare Practitioners (HCPs) holding powerful positions in decision-making bodies, involvement in research, considerations of reputation and career or sponsorship from companies.

… they might be offered honorarium or stipends, or a range of services, such as ghostwriters for medical journals, which will indirectly enhance or directly enhance their professional standing and from that flows their ability to garner research funding and have high profile research teams, so one can very directly benefit… even if the money isn't coming to you if it's coming to your research centre, it gives some people a lot of influence… (P16, patient with long term condition)

Conflicts could also be caused by the way healthcare systems were organised and funded. Levers within the NHS could include GPs prescribing generically to save money, incentives to promote a product or 'up-selling' on top of routine care. These could be regarded as a conflict, particularly when there was an uncertain justification.

do I need anti-glare on my glasses? I don't really know. But the lovely young person tells me that it's the best thing ever. So…the transaction becomes more complicated. Because there's a commercial element to the transaction that's being played out. (P19, lay, routine appointment for eyesight check at optician)

These conflicts could be unforeseen, and related to influence and power. For instance, professionals' involvement in research was a potential conflict, as interest in certain outcomes could lead to biassed treatment recommendations.

Well, it means that the prescriber or the provider of the service is making decisions that aren't just in the interest of the patient or the user. That they may give too much priority to their own interests. Those might not be financial. They might be their research project, or something that they're supporting. But it's not putting the patient first, second, third and fourth, as it were. (P4, patient)

Participants expressed concerns that professionals may lack insight into their conflicts or their potential impact, due to unclear definitions, the lack of oversight of COIs or professionals' unconscious bias. Participants thought that peer review, and training on how to recognise COIs could potentially help. Respondents also argued that personal beliefs may result in unconscious bias; intellectual bias was also viewed as damaging. One participant felt that HCPs do not understand the bias in sciences generally and therefore do not recognise the role of reflexivity when evidence is scrutinised to inform treatment decisions.

So, I actually think healthcare professionals don't understand what conflict of interest means…They don't understand biases in science. They don't understand their need for not only reflection, but reflexivity. That they need to reflect on their own values and beliefs in what they're bringing to the evidence that they're presenting to patients. (P7, patient activist)

### DOI in healthcare can be hard to find

While participants were not asked directly about trust and transparency, patient activists and representatives repeatedly expressed concern about how the difficulty finding declarations could lead to a loss of trust in the medical profession.

I think that in that last scenario, the reason why my trust diminishes is because of lack of transparency. I think that is why would you not tell someone?…Why would you not tell someone that you have a financial interest in this? So I suppose it's the discovery—you feel like you've been duped. (P10, activist)

For some, transparency was a tool to navigate the vague nature of COIs, enabling patients to make decisions and reinforce the trust in HCPs. Equally, participants highlighted their desire for and expectation of trust in the medical profession. Undisclosed COIs could lead to potential or actual loss of trust, particularly when conflicts were undisclosed.

I'd like it to be easier…. and I may not have found all of them. (P13, lay, patient representative roles)

### COIs present both challenges and benefits for patients

None of the participants questioned the value of research. However, contradictory aspects of COIs were raised by participants in relation to the interplay between industry and the profession. Some participants thought that industry funding for education might be worth accepting for their potential benefits. A small number (who also represented participants who had worked in the pharmaceutical industry) highlighted the positive role of industry/the private sector in providing education and training to trainees and HCPs. One participant described this as justified:

yes, we need to have the declaration of a conflict of interest, but we actually have to allow a little bit of a conflict of interest for them to get to congresses and get educated…because if they don't, as I say, it's to our detriment ultimately, I believe… (P17, activist, experience working in industry)

Others recognised the potential for industry sponsored education to be a potential conflict (professionals obtaining free education that would otherwise have to be

paid for) and biassed in nature, leading to poorer quality healthcare.

…One assumes that they were being promoted by the company at the time. The vaginal mesh incidence, again, one assumes across the gynaecological board, they were being hyped as the best thing. And that has ruined people's lives, in fact, it's killed some people. (P9, lay)

Complexity was described. COIs (such as sponsored education) could lead to improved care, but the presence of a conflict could lead to an assumption that decisions were not in the best interest of the patient—whether or not they were. Two participants described suffering harm because of COIs around the treatment they received. Others had heard about such experiences or felt that their health needs were not always the primary consideration of the HCP because of conflicts. They discussed how COIs can lead to corruption and poor patient care by referring to the difficulties in defining and acknowledging COIs and the role of the trust in patient—HCP relationship.

I like to start from the point that people are trustworthy and doing things from a good motive… There's a scenario where that doctor is somebody who genuinely wants to help patients, genuinely believes in the product, genuinely thinks that it's the right product for me, and happens to have received some payment for his role in developing it because his expertise has a value. (P1, patient)

### COIs need to be carefully managed

Participants overall described COIs as challenging to manage, citing a combination of difficult definitions, variable significance and the different amounts of information patients were felt likely to want. The potential for information overload for patients, complicating decision-making, was discussed. Some patients described the practical difficulties of HCP making disclosures in a time-limited consultation, and the burden then put on patients to effectively consider and/or manage them.

I can also see that people, perhaps, in a consultation, are overwhelmed with information about their health… And it's so very difficult to know how relevant it is and whether it's really something that is swaying their judgement or not. (P6, citizen)

Concerns were expressed about workload for HCPs if more regulations were introduced, with extra time needed for disclosure and explanation rather than for direct patient care.

And also, these professionals are very precious to you, they don't have much time, and you don't want to be talking about conflicts of interest when you actually want them to help you to do what it is that they're going to do. (P5, activist)

Participants suggested ways of managing COIs, including better transparency. Several participants suggested that a mandatory register or regulation, like a Sunshine Act[6] might be a useful way of managing COIs. Respondents also wanted to see the same rules applied across the whole NHS as variation across the health boards was seen to lead to fragmentation and variation in care.

I think there should be… This Sunshine Act in America they have to declare… And I thought well that's open to interpretation but at least it would be something, so it would make it more ethical. (P12, lay)

However, systems were needed to deal with COIs in ways that accounted for medical power. Complexity was again reflected. Participants wondered if simple disclosure would make a difference to patients. For example:

Most patients won't ever question this. They will never question their doctors. (P2, lay)

A view held by several participants was that managing COIs should not rest with patients, but be a professional duty. The multiplicity of interactions meant that patients should be protected because a simple statement of interests could not suffice.

But does that capture the reality of the communications between doctor and patient, where so much is non-verbal, or implicit? …Relationships with employer, relationship with fellow professionals, relationship with marketing reps and others. The more you look at it, the harder it gets, I found. (P3, lay)

Participants felt that independent advocacy could aid vulnerable people in consenting to treatment, if there were concerns around COIs. Second opinions and the presence of independent advocates were also seen as ways to mitigate bias. Independent parties would be able to raise issues of concern in relation to COIs:

I think I would need an advocate with me to discuss my options. So, that's somebody who understands the medical side, but also somebody who is trained in advocacy and is a patient advocate… (P21, lay)

However, again, complexity was acknowledged as independence was not always guaranteed.

I work in the area of health communication….the challenge I'm finding is that most of the patient advocates are sponsored by drug companies. And they've no regulation around that. There's no guidelines… it just seems a bit murky to me and not transparent. (P7, activist)

Some participants highlighted where a declaration of a conflict may not necessarily reveal a negative bias. This could lead to patient concern about a conflict that was not in fact significant.

that doesn't mean somebody's not a good person and they're not doing the right thing…you know, he really believes in all the stuff…But he must be biassed by that. Is that good bias? Is that bad bias? I don't know, you know?… (P15, carer)

Participants suggested 'spot checks' (P17, activist) or oversight by an independent body to ensure that HCPs do not submit unsubstantiated information. This contrasted with discussions around trying to limit additional bureaucracy. Checking, peer review, and training were suggested on how to recognise COIs and potentially reduce unconscious bias. Other professions, particularly the public sector, law, parliament and academia were compared, reflecting a shared view that professionals had duties to organise effective management.

I think the way it could be solved is that doctors with direct financial interests don't take on cases in that particular area. Obviously, that's quite serious, but I can't think of any other way that it could be totally solved. You know, essentially like the legal profession, people recuse themselves from cases. So that's just essentially the same principle. (P18, citizen)

## DISCUSSION

This is the first study we are aware of to test whether DOI, as currently made by UK based, publicly facing doctors, could be located by laypeople and professionals. In the second phase of the study, in vignettes describing an increase in financial declarations, participants' confidence in the doctor acting in their best interests, and their trust, fell. While 85% of participants thought that knowing about professional declarations was 'definitely' or 'probably important', despite generous marking, 27% (58) participants found a relevant link to 'fit for purpose' information and only 2% found all components of the 'gold standard'.

All the professionals taking part in the study were part of an organisation with an official need to publicly declare interests. This study has demonstrated that, despite the efforts of the NHS to improve practice (eg, NHS Trusts holding public registers of interest), transparency is not being effectively achieved because participants were unable to locate the registers with reasonable ease.

The third phase of this study invited patients and citizens, who had participated in the second phase, to interview. Declarations and conflicts presented complex challenges. Participants described ways to improve the system, but were also concerned for practicality, opportunity cost and bureaucracy. This was particularly in terms of where declarations should be made, when they were relevant, and how patients could feasibly use these. Multiple trade-offs were described. For example, free but sponsored education and training potentially resulted in professionals having more up-to-date knowledge but COI could be produced by these funders. Medical power had

to be mitigated. Professionals may lack insight into their own bias, and independent oversight would be necessary to mitigate and check compliance. Some expressed concern of the additional burden that could be placed on doctors, and also patients to locate and interpret declarations. Free-text responses in the searches confirmed this, for example 'Despite searching I found no good way to find any', 'I could not easily find anything apart from some stuff on Wikipedia', "Sorry, I struggle to find any. Giving up.' and 'I am finding this task really difficult. There is too much information to work through. Patients should not have to do this research.'

Strengths of this study include the collaboration of a patient panel to develop the survey, suggest professional participants and disseminate invitations to participate. Additionally, it was highly pragmatic, mimicking the steps that a citizen would take to investigate a conflict. This is the first study we are aware of which tested current UK declaration processes. It is also the first we are aware of in the UK which interviewed laypeople to elucidate understanding and concerns about COI in healthcare professionals, and what improvements could consist of. There were several limitations. The HPP were by definition in the public eye, and willing to take part, and may have been more likely to use a high profile register, for example, on government websites. However this would have resulted in an overestimation of being able to locate conflicts, meaning that the results would be artificially high. Only one conflict in each category (eg, professional financial, indirect) had to be found to be scored correctly, leading to an overestimation of the effectiveness of current practice. Declarations were scored equally, however, the relative importance of each declaration is not, in reality, equal: some may have been unimportant and very unlikely to cause important conflict; others, the opposite. Further, the participants and professionals who took part in our study are likely atypical, with engagement with these issues prior to the request for participation. It is not expected that many citizens would normally spontaneously search for healthcare professionals' declarations or conflicts. Our participants are likely to have engaged with some or many of the issues related to COI, given that our patient group assisted in sending the questionnaire widely to engaged patient groups. Multiple entries to the questionnaire by one person under different email addresses would have been possible but we consider this overall unlikely to have had a large impact on results. Nevertheless, even in a group of activated professionals and citizens, finding a complete DOI was extremely difficult, and a partial finding of declarations was possible only a minority of the time. Participants agreed, reporting a low level of confidence that their results were complete.

While there are no directly comparable studies, other US work[28] supports the finding that patients, including potential research participants, wish transparency and to know the researchers' COI. The impact of disclosure of a doctor's COI to patients via a mailed letter has been investigated in the USA.[18] Patients subsequently

described an increased level of confidence in their ability to judge potential impacts of conflicts on their healthcare. Overall there was no change in described levels of trust in doctors. However, in patients who 3 months later recalled receiving the disclosure notice, around one-fifth described increased trust in the physician. Given what is known about the potential for harm from financial COI, it is questionable whether this is a good outcome, as people may trust advice even if it is at risk of bias. Sah et al[29] investigated trust in scenarios, mainly concerning financial advice, where the interests of client and advisor were aligned or not. Trust was found to be reduced when a conflict of interest was known, even when the interest of client and advisor was congruent and the advice was high quality. The final scenario concerned a medical vignette. This found that a disclosure of a conflict resulted in increased trust in participants. However the vignette featured a doctor's recommendation not to do a test, which would otherwise have attracted a fee. This so-called 'altruistic signal' is theorised to offset the 'disclosure penalty' which can otherwise reduce trust. Finally, a field experiment in the USA[30] randomised patients to receive a hospital appointment letter containing, or not, the doctor's conflict of interest statement. Patients receiving the disclosure reported more knowledge of these conflicts, with no change in trust or appointment attendances. These studies were set in the USA, where there are major cultural differences concerning health service delivery, and are unlikely to be directly applicable to dissimilar countries such as the UK.

This research has found that laypeople hold mixed and often nuanced views over COI. Further, the practical aspects of declarations, including organisation, workload for both patients and doctors, and interpretation, was realised to present difficulties. Given the strong decline in trust in the vignettes with increasing COI, it is uncertain whether an 'altruistic signal' would compensate for a 'disclosure penalty'. Further, the 'medical power' which laypeople alluded to must still be negotiated where conflicts are found, an aspect not investigated in these other studies, although Pearson et al[18] found that more than half the patients who remembered seeing a disclosure did not feel they knew enough to judge the potential impact of it.

This leads to basic questions about the use and purpose of DOI in the UK. Should they be for transparency alone? Should declarations be intended as more than an 'information dump' but made in ways which enable judgements—and effective management?

Unintended consequences of transparency are possible, for example, 'moral licence', where disclosure is assumed to negate potential bias. There is evidence that doctors believe that other doctors become biassed when exposed to small gifts, while they themselves do not.[31] Further, the survey and interviews confirmed that a lack of transparency in professionals leads to less trust for patients. If more transparency was created, and trust in conflicted doctors increased, it is uncertain whether this would be

justified, given the evidence that financial COI are associated with bias, and more expensive poorer quality healthcare.[32 33] Indeed, previous research has found that DOI, for example, those recommended by the International Committee of Medical Journal Editors, are of poor quality and make it difficult for the reader to assess bias.[34]

Transparency may therefore not be a benign act. Given that most UK Hospital Trusts do not state the action planned to mitigate a publicly declared conflict, it may not be clear where the work of finding, interpreting and managing COIs rests.[21] Many respondents were concerned as to what to do with the conflicts located. Some felt they could trust the doctor regardless of a conflict, as disclosure mitigated bias; others felt it difficult to know whether they could trust the doctor's judgement despite a declared conflict, and would require advocacy to assist. While patient organisations were suggested as potential advocates, there is also evidence that some are themselves conflicted.[35]

The General Medical Council (UK) recommends doctors 'avoid conflicts of interest wherever possible' and 'declare any conflict to anyone affected, formally and as early as possible, in line with the policies of your employer or the organisation contracting your services'. They also recognise 'Conflicts of interest are not always avoidable… follow(ing) established procedures for declaring and managing a conflict'.[13] The risk is that disclosure is used to include rather than exclude individuals from relevant decision-making, in the belief that a disclosure constitutes management. While disclosure is necessary for management, it does not substitute for it. This is particularly important when considering the evidence on unconscious bias from professionals, and the risk of trusting conflicted, but declaring, professionals where patients have limited power to know about or mitigate the potential impact of a conflict.

Further research should elucidate what the purpose(s) of DOI should be for different groups of people and find ways to meet these needs. For example, if it is for simple transparency, declarations need to be easy to find and understand. If they are to manage conflicts, a clear decision on boundaries may be helpful. However, changes should be tested not just with patients, but professionals, as issues of workload and opportunity cost were reflected in interviews. Further, research should help to understand the best ways of making declarations which allows the reader to make an evidence-based interpretation of their potential impact. Our study relied on interested parties to participate, and research on panels recruited from the wider population would be helpful. Professional views should also be sought to understand what the facilitators and barriers are to making declarations in order to organise the best way to declare and manage them. Finally, the value of making declarations has yet to be established, beyond transparency. A more reliable way to manage them may be via better processes

of disallowing certain conflicts from defined roles, rather than trying to manage them using haphazard DOI. Further qualitative work may help to understand how conflicts are managed in practice.

## CONCLUSION

DOI by professionals are agreed to be important, but are unfit for purpose in their current form. The survey found that patients describe trusting professionals with no conflicts the most, and professionals with undisclosed conflicts the least. The practical task of finding DOI for well-known doctors in the public domain was difficult. Even when disclosure statements were found, most were incomplete when compared with the 'gold standard', which were rarely located. Interviews with laypeople found nuanced views about disclosure and management of conflicts. They were described as important, but difficult to find and use, and although some potential conflicts could be justified, they needed to be managed. Currently practitioners making declarations cannot be assured that this information can be readily found, and cannot assume that this information can be used in decision-making by laypeople. Other research finds that transparency may result in unintended consequences, including placing trust in professionals who may take 'moral licence' from an open declaration, while increasing workload for patients and professionals. Together, these may cause unintended harms. Patients may not feel able to use information about COI to their advantage. This means that declaring potential conflicts should be refined, with greater professional emphasis of avoiding, identifying and managing serious COI and clear, public definitions on who requires exclusion from what types of decision making. It is questioned whether the purpose of declarations should rest on providing mere transparency, but be used to exclude, rather than include, conflicted professionals in relevant decision-making.

### Checklist

We enclose the Consensus Based Checklist for Reporting of Study Results (CROSS) (for survey designs) and the Standards for Reporting Qualatative Research (SRQR) (for qualitative research) checklists as recommended by EQUATOR.

**Acknowledgements** We thank the health professionals who took part in this study for their time and valuable assistance and support; our patient panel, Lynne Craven, Brian Finney, Fiona Raje and Kath Samson, for their help with study design, invitations and recruitment, and Professor Carl Heneghan who discussed and advised on the study at conception. We also thank the peer reviewers for their constructive and helpful comments.

**Contributors** Study conceived by MMC with design by MMC and FS. Questionnaire design by MMC, RM, GO and A-MB with assistance and direction from the patient panel. Analysis of questionnaire and survey data by RM, MMC and FS. Qualitative interviews by KM and MMC. Analysis of interviews by KM, A-MB, GO and MMC. Draft of paper by MMC with editing and contributions from all. MM is the guarantor.

**Funding** MM is grateful to the Chief Scientist Office Scotland for funding an NRS Career Researcher Fellowship. This study was otherwise funded by University of St Andrews Research Fund. MM's research position is funded by the Chief Scientist Office in Scotland.

**Competing interests** MMC has been paid for writing/broadcasting in relation to issues connected with campaigning around conflicts of interest. All authors have completed the Unified Competing Interest form (available on request from the corresponding author) and have no other declarations to make regarding support from any organisation for the submitted work; financial relationships with any organisations that might have an interest in the submitted work in the previous 3 years.

**Patient and public involvement** Patients and/or the public were involved in the design, or conduct, or reporting, or dissemination plans of this research. Refer to the Methods section for further details.

**Patient consent for publication** Not applicable.

**Ethics approval** This project was approved by the University of St Andrews School of Medicine Ethics Committee, MD16045. Participants gave informed consent to participate in the study before taking part.

**Provenance and peer review** Not commissioned; externally peer reviewed.

**Data availability statement** No data are available. No further data is available. This is because our health professional participant gave consent on the basis of confidentiality of their identity, which would not be possible if further data was made public.

**ORCID iDs**
Margaret McCartney http://orcid.org/0000-0002-7238-6358
Katrin Metsis http://orcid.org/0000-0003-0827-6557
Ronald MacDonald http://orcid.org/0000-0002-6197-5675
Frank Sullivan http://orcid.org/0000-0002-6623-4964
Gozde Ozakinci http://orcid.org/0000-0001-5869-3274
Anne-Marie Boylan http://orcid.org/0000-0001-8187-0742

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
