## [Reviewer comments · BMJ Open]

ARTICLE DETAILS

TITLE (PROVISIONAL)	You feel like you've been duped". Is the current system for health professionals declaring potential conflicts of interest in the UK fit for purpose? A mixed methods study.
AUTHORS	McCartney, Margaret; Metsis, Katrin; MacDonald, Ronald; Sullivan, Frank; Ozakinci, Gozde; Boylan, Anne-Marie

VERSION 1 – REVIEW

REVIEWER	Parker, Lisa The University of Sydney, Centre for Values, Ethics and the Law in Medicine
REVIEW RETURNED	09-Mar-2023

GENERAL COMMENTS	Comments to the Authors This revised version of the paper is much clearer and I commend the authors for their reworking. I have some minor comments as per below. I would encourage the authors to take more care with their terminology, for example p4, line 50 they write about "potential coi" and again on p 10 line 19 they write "professionals' involvement in research was a potential conflict". Perhaps this is a personal use of words, but since coi is defined as 'a risk' of bias/influence from a secondary interest, I would suggest that the word 'potential' [conflict] is unnecessary. Another terminology check is on p 13, line 52 where the authors write "bias and/or conflicts of interest could be produced by these funders" – bias and coi are not interchangeable, rather (as the authors discuss in the Intro) coi is a situation where there is a risk of bias. I pick up on these small but important terminology issues because I think inconsistent terminology contributes to misunderstanding in this field ! P4 line 52 the provided definition of financial coi is more limited than my reading would suggest – the authors write that financial coi is 'where an individual receives or may receive a direct financial benefit from the consequences of awarding of funding' with an example being financial benefit from increases sales in a product. For the purposes of medical professionals, financial coi is widely held to include multiple sources of industry money – as listed by the authors on p 7, line 56-57 eg for travel to conferences, to cover research costs, for medical advisory services, for speaking at conferences or events. P5 line 7 The authors skim briefly over the purpose of physician
--

	declaration of coi beyond patient knowledge. From my reading of the literature, patient knowledge is just one reason for physicians to declare coi. I acknowledge that patient / lay person knowledge and interpretation is the focus of this paper, but suggest the authors expand on other important reasons to require coi declaration eg to identify and then ban physicians with coi from guideline committee chair or membership; identify and ban or limit conflicted physicians from medical student teaching; to assist readers interpret clinical studies and/or academic opinion pieces written by conflicted experts; to encourage physicians to rethink/choose to avoid industry money so as to avoid being on public registers. I wonder whether this should be discussed in more detail in the Intro to set up the comments in the Discussion and particularly Conclusion about possible unintended consequences from transparency and other purposes of declarations of coi. Otherwise the comments in the Conclusion about other purposes of coi declarations seem to come out of nowhere. I have written about this topic in the past and you might find some useful readings in my ref list Parker L, Karanges EA, Bero L. Changes in the type and amount of spending disclosed by Australian pharmaceutical companies: an observational study. BMJ Open 2019;9:e024928. doi:10.1136/ bmjopen-2018-024928 P 10 lines 10-14 this quote is the same as one used on the previous page. It's not clear how the duplication of this quote is useful or necessary.
--	---

VERSION 1 – AUTHOR RESPONSE

Reviewer: 1

Dr. Lisa Parker, The University of Sydney

Comments to the Author:

BMJ Open _duped revised

Comments to the Authors

This revised version of the paper is much clearer and I commend the authors for their reworking. I have some minor comments as per below.

I would encourage the authors to take more care with their terminology, for example p4, line 50 they write about “potential coi” and again on p 10 line 19 they write “professionals ’involvement in research was a potential conflict”. Perhaps this is a personal use of words, but since coi is defined as ‘a risk ’of bias/influence from a secondary interest, I would suggest that the word ‘potential ’[conflict] is unnecessary.

- Our use of language is intended to be clear for the generalist medical reader who may not be aware of this interpretation. We reflect the International Committee of Medical Journal Editors and NHS wording. For example, the ICMJE says “The potential for conflict of interest and bias exists when professional judgment concerning a primary interest (such as patients' welfare or the validity of research) may be influenced by a secondary interest (such as financial gain). Perceptions of conflict of interest are as important as actual conflicts of interest ” (<https://www.icmje.org/recommendations/browse/roles-and-responsibilities/author-responsibilities--conflicts-of-interest.html>). NHS England says “...a ‘conflict of interest ’is defined a: “A set of circumstances by which a reasonable person would consider that an individual’s ability to apply judgement or act, in the context of delivering, commissioning, or assuring taxpayer funded health and care services is, or could be, impaired or influenced by another interest they

hold.” <https://www.england.nhs.uk/wp-content/uploads/2017/02/guidance-managing-conflicts-of-interest-nhs.pdf>. In Scotland, the NHS National Services fraud prevention guide says that “A conflict of interest is a situation in which an individual has competing interests or loyalties”. <https://www.nss.nhs.scot/countering-fraud/prevention/short-fraud-guides/procurement-fraud/> In other words, we do not think that it is widely (internationally) accepted that a description of a “conflict of interest” includes a potential for conflict, by definition. Our language reflects this ambiguity. For example, there may be an appearance of a COI, but not a proven COI; judgements on when an interest represents a conflict of interest may vary.

Another terminology check is on p 13, line 52 where the authors write “bias and/or conflicts of interest could be produced by these funders” – bias and coi are not interchangeable, rather (as the authors discuss in the Intro) coi is a situation where there is a risk of bias. I pick up on these small but important terminology issues because I think inconsistent terminology contributes to misunderstanding in this field !

- In the examples given, we reflect the ambiguity of the terms in common use. We appreciate that bias is not the same as COI, and we have removed the word ‘bias’.

P4 line 52 the provided definition of financial coi is more limited than my reading would suggest – the authors write that financial coi is ‘where an individual receives or may receive a direct financial benefit from the consequences of awarding of funding ’with an example being financial benefit from increases sales in a product. For the purposes of medical professionals, financial coi is widely held to include multiple sources of industry money – as listed by the authors on p 7, line 56-57 eg for travel to conferences, to cover research costs, for medical advisory services, for speaking at conferences or events.

- We agree, and this list was not meant to be a definitive list, but rather a summary of what NHS England gave as examples. For the avoidance of doubt, we have expanded the list of examples.

P5 line 7 The authors skim briefly over the purpose of physician declaration of coi beyond patient knowledge. From my reading of the literature, patient knowledge is just one reason for physicians to declare coi. I acknowledge that patient / lay person knowledge and interpretation is the focus of this paper, but suggest the authors expand on other important reasons to require coi declaration eg to identify and then ban physicians with coi from guideline committee chair or membership; identify and ban or limit conflicted physicians from medical student teaching; to assist readers interpret clinical studies and/or academic opinion pieces written by conflicted experts; to encourage physicians to rethink/choose to avoid industry money so as to avoid being on public registers. I wonder whether this should be discussed in more detail in the Intro to set up the comments in the Discussion and particularly Conclusion about possible unintended consequences from transparency and other purposes of declarations of coi. Otherwise the comments in the Conclusion about other purposes of coi declarations seem to come out of nowhere. I have written about this topic in the past and you might find some useful readings in my ref list Parker L, Karanges EA, Bero L. Changes in the type and amount of spending disclosed by Australian pharmaceutical companies: an observational study. *BMJ Open* 2019;9:e024928. doi:10.1136/ bmjopen-2018-024928

- Many thanks. This was indeed the focus of our paper. We did discuss other reasons for declarations beyond patient knowledge, for example : “Declarations of interest may be intended to direct their management, for example, in guideline committees (12); regulators may also stipulate specific conflicts which must be declared to patients (13). This has also been emphasised in the IMMDS review, which has called for patients to be able to access declarations in order to “reach informed decisions about who is best to treat us” (7). There are therefore multiple purposes to which declarations can be put.” We do however agree that it is useful to expand this. We have done so and think this improves the paper and the validity of the conclusions ,thank you.

P 10 lines 10-14 this quote is the same as one used on the previous page. It’s not clear how the duplication of this quote is useful or necessary

- Many thanks - this was an error which has been rectified. We appreciate this point.

VERSION 2 – REVIEW

REVIEWER	Parker, Lisa The University of Sydney, Centre for Values, Ethics and the Law in Medicine
REVIEW RETURNED	26-May-2023
GENERAL COMMENTS	The authors have address my comments adequately. I look forward to publication of this paper.